# Amination of 5-Spiro-Substituted 3-Hydroxy-1,5-dihydro-2*H*-pyrrol-2-ones

**DOI:** 10.3390/molecules26237179

**Published:** 2021-11-26

**Authors:** Ekaterina E. Khramtsova, Ekaterina A. Lystsova, Evgeniya V. Khokhlova, Maksim V. Dmitriev, Andrey N. Maslivets

**Affiliations:** Department of Chemistry, Perm State University, ul. Bukireva 15, 614990 Perm, Russia; liscova_ea@mail.ru (E.A.L.); masliv2@psu.ru (E.V.K.); maxperm@yandex.ru (M.V.D.); koh2@psu.ru (A.N.M.)

**Keywords:** amination, carbodiimide, pyrrole-2-one, thermolysis, urea

## Abstract

The 3-hydroxy-1,5-dihydro-2*H*-pyrrol-2-one motif is a valuable scaffold in drug discovery. The replacement of the 3-oxy fragment in 3-hydroxy-1,5-dihydro-2*H*-pyrrol-2-ones-based compounds with a 3-amino one (3-amino analogs of 3-hydroxy-1,5-dihydro-2*H*-pyrrol-2-ones, 3-amino-1,5-dihydro-2*H*-pyrrol-2-ones) can play a crucial role in their biological effect. Thus, approaches to 3-amino-1,5-dihydro-2*H*-pyrrol-2-ones are of significant interest. We developed an approach to 5-spiro-substituted 3-amino-1,5-dihydro-2*H*-pyrrol-2-ones that could not be obtained using previously reported approaches (reactions of 3-hydroxy-1,5-dihydro-2*H*-pyrrol-2-ones with amines). The developed approach is based on the thermal decomposition of 1,3-disubstituted urea derivatives of 5-spiro-substituted 3-hydroxy-1,5-dihydro-2*H*-pyrrol-2-ones, which were prepared *via* their reaction with carbodiimides.

## 1. Introduction

The 3-hydroxy-1,5-dihydro-2*H*-pyrrol-2-one (HDP) motif (Figure 1) is a valuable scaffold in drug discovery. Inhibitors of p53–MDM2 protein–protein interaction [1,2], inhibitors of HIV integrase [3,4], antibacterial agents against methicillin-resistant *Staphylococcus aureus* [5], dengue virus helicase inhibitors [6], P2X3 receptor antagonists [7], and other potential pharmaceuticals [8,9] have been developed on its basis.

The replacement of the 3-oxy fragment in HDP-based compounds with a 3-amino one (3-amino analogs of HDPs, 3-amino-1,5-dihydro-2*H*-pyrrol-2-ones (ADPs)) can play a crucial role in their biological effect (Figure 1) [1,2,9]. Thus, to investigate a wider chemical space around the HDP scaffold, new approaches to ADPs are of significant interest.

There are three universal strategies towards ADPs enabling their synthesis with a wide range of substituents (Figure 1). These strategies are the substitution of hydroxyl groups in the corresponding HDPs *via* their reactions with amines (pathway *a*) [1,2,7,9,10,11,12], the multicomponent reactions of amines and aldehydes with pyruvic acid derivatives (pathway *b*) [13,14] or acetylenedicarboxylates (pathway *c*) [15,16] (Figure 1).

As a part of our long-standing interest in the synthesis of spiro compounds bearing the HDP moiety [17], we wanted to prepare their 3-amino analogs. However, we were unable to synthesize the desired compounds from HDPs and amines [1,2,7,9,10,11,12], and therefore, herein, we report an alternative approach to ADPs (Figure 2).

## 2. Results and Discussion

Initially, we prepared HDPs bearing a rigid spiro substituent at the C^5^ position, compounds **1a–g**, by the reaction of compounds **2a–g** with dicyclohexylurea (Figure 3) [17,18,19]. This reaction proceeded smoothly and yielded target compounds **1a–g** in excellent yields (isolated yields of 86–98%). The spectral data of compounds **1a–g** were in good agreement with other similar spiro HDPs reported by our group earlier [18,19]. Compounds **1a–g** were isolated by simple filtration from the reaction mixture and were sufficiently pure to be used in further experiments without additional purification.

Then, we performed a series of reactions of compound **1a** with cyclohexylamine (Table 1) in order to obtain the required cyclohexylamino derivative **3a**. The examined conditions (Table 1) were selected according to the previously reported substitution reactions of hydroxyl groups in various non-spiro HDPs (bearing hydrogen and a substituent at the C*^5^* position) with amines [1,2,7,9,10,11,12]. Although the examined conditions were productive in the reported cases [1,2,7,9,10,11,12], in our study, none of them led to the desired results (Table 1). We observed that the reaction mixtures contained unconverted compound **1a** (93–99%), cyclohexylamine and small quantities of unidentified side products. The target compound **3a** was only observed in trace amounts when 1,4-dioxane was used as the solvent (Table 1, Entries 10,11).

We suppose that such inaction of compound **1a** in the reaction with cyclohexylamine could be explained by the fact that the desired transformation proceeded *via* the participation of the corresponding HDP **A** in keto form, which underwent a condensation reaction with amine to afford the target 3-amino derivative of HDP **B** (Figure 4) [12]. Compound **1a** (Figure 4, structure **A**, R^2^ + R^3^ = spiro) had less conformational flexibility and less propensity for tautomerization to its keto form due to the influence of the rigid spiro-substituent at the C^5^ position of the pyrrol-2-one moiety than the previously reported [1,2,7,9,10,11,12] non-spiro HDPs bearing hydrogen and a substituent at the C^5^ position (Figure 4, structure **A**, R^2^ = H, R^3^ = Alk, Ar, H).

Therefore, we decided to develop a new strategy based on the ability of urea and alkyl/arylureas to decompose with the formation of ammonia/amines and isocyanates (Figure 5) [20,21].

For the implementation of our strategy, we obtained compounds **4a,b**, 1,3-dialkylurea derivatives of compound **1a**, *via* the reaction of compound **1a** with inexpensive and commercially available dialkylcarbodiimides **5a,b** (Figure 6). The reaction with dicyclohexylcarbodiimide (DCC) **5a** proceeded smoothly at a reagent ratio of 1:1 to give the target compound **4a** in an excellent yield (93%). However, the reaction with diisopropylcarbodiimide (DIC) **5b** at a reagent ratio of 1:1 yielded the target compound **4b** in a lower yield (73%). Such a decrease, when changing DCC to DIC, could be because the boiling point of DIC (145–146 °C) [22] is lower than DCC (151–152 °C at 10 mmHg) [22], and DIC partially evaporated from the reaction mixture. Increasing the DIC ratio to two equivalents increased the isolated yield of compound **4b** to 88%. Compounds **4a,b** were isolated by filtration from the reaction mixture and were sufficiently pure to be used in further experiments without additional purification. Then, the proposed method was successfully implemented to the synthesis of compounds **4c–n** (Figure 6). This approach worked well with various aryl substituents in compounds **1** (Figure 6). The involvement of aryl and primary alkyl-bearing carbodiimides **5c**,**d** in this reaction was successful too (Figure 6).

Probably, the formation of compounds **4a–n** proceeded through the pathway of acyl transfer in isourea intermediate **C** (Figure 7), which is a common concurrent process for the Steglich esterification [23]. It should be emphasized that although compounds **1a–g** contain two different hydroxy groups, enolic and phenolic ones, the reaction with carbodiimides **5a–d** proceeded exclusively at the enolic hydroxy group, which could be explained by its higher acidity in comparison with the phenolic one, and, consequently, higher reactivity in reactions with carbodiimides [24].

Having synthesized derivatives **4a–n**, we investigated the possibility of their thermal decomposition to afford the desired amino derivatives of HDPs, compounds **3a–n** (Table 2). We found that compounds **4a–n** readily decomposed at their melting point temperatures to afford the desired compounds **3a–n** (84–98%). The proposed method was suitable for various aroyl substituents and R’ substituents in a urea moiety of compounds **4a–n** (Table 2).

The structures of compounds **3b**, **3c**, and **4h** were proved by single-crystal X-ray analyses (CCDC 2090985 (**3b**), 2090984 (**3c**), 2090986 (**4h**)).

Preliminary antimicrobial assays [25] of compounds **1a–g** and **4a,c–h** were carried out (detailed data are given in Supporting Materials). Unfortunately, we found that the tested compounds did not show any significant antimicrobial activity (against *Staphylococcus aureus*, *Escherichia coli*, *Klebsiella pneumoniae*, *Acinetobacter baumannii*, *Pseudomonas aeruginosa*, *Candida albicans*, *Cryptococcus neoformans var. grubii*) in vitro.

## 3. Materials and Methods

### 3.1. General Information

^1^H and ^13^C NMR spectra (Appendix A) were acquired on a Bruker Avance III 400 HD spectrometer (Bruker BioSpin AG, Fällanden, Switzerland) (at 400 and 100 MHz, respectively) in CDCl_3_ (stab. with Ag) or DMSO-*d*_6_ using the HMDS signal (in ^1^H NMR) or solvent residual signals (in ^13^C NMR, 77.00 for CDCl_3_, 39.51 for DMSO-*d*_6_; in ^1^H NMR, 7.26 for CDCl_3_, 2.50 for DMSO-*d*_6_) as internal standards. IR spectra were recorded on a Perkin–Elmer Spectrum Two spectrometer (PerkinElmer, Waltham, MA, USA) from mulls in mineral oil. Melting points were measured on a Khimlabpribor PTP apparatus (Pribor-T, Saratov, Russia) or a Mettler Toledo MP70 apparatus (Mettler-Toledo GmbH, Greifensee, Switzerland). Elemental analyses were carried out on a Vario MICRO Cube analyzer (Elementar, Langenselbold, Germany). The reaction conditions were optimized using UPLC-UV-MS (Waters ACQUITY UPLC I-Class system (USA); Acquity UPLC BEH C18 column, grain size of 1.7 μm; acetonitrile–water as eluents; flow rate of 0.6 mL/min; ACQUITY UPLC PDA eλ Detector (Thermo Fisher Scientific, Waltham, MA, USA) (wavelength range of 230–780 nm); Xevo TQD mass detector (Agilent, Santa Clara, CA, USA); electrospray ionization (ESI); positive and negative ion detection; ion source temperature of 150 °C; capillary voltage of 3500–4000 V; cone voltage of 20–70 V; vaporizer temperature of 200 °C). The single-crystal X-ray analyses of compounds **3b**, **3c**, and **4h** were performed on an Xcalibur Ruby diffractometer (Agilent Technologies, Cheadle, UK). The empirical absorption correction was introduced by the multi-scan method using SCALE3 ABSPACK algorithm [26]. Using OLEX2 [27], the structures were solved with the SHELXS-97 [28] program and refined by the full-matrix least-squares minimization in the anisotropic approximation for all non-hydrogen atoms with the SHELXL (accessed on 1 March 2018) [29] program. Hydrogen atoms bound to carbon were positioned geometrically and refined using a riding model. The hydrogen atoms of OH and NH groups were refined freely with isotropic displacement parameters. The contribution of the solvent electron density (for compounds **3b** and **4h**) was removed using the SQUEEZE routine in PLATON [30]. Thin-layer chromatography (TLC) was performed on Merck silica gel 60 F_254_ plates using EtOAc/toluene, 1:5 *v/v*, toluene, EtOAc as eluents. Starting compounds **2a–g** were obtained according to reported procedures [31] from oxalyl chloride (purchased from commercial vendors) and heterocyclic enamines (obtained according to reported procedures [31] from commercially available reagents). Toluene for procedures involving compounds **2a–g** was dried over Na before use. All other solvents and reagents were purchased from commercial vendors and used as received. Procedures involving compounds **2a–g** were carried out in oven-dried glassware.

### 3.2. Synthetic Methods and Analytic Data of Compounds

#### 3.2.1. General Procedure to Compounds **1a–g**

A suspension of the corresponding compound **2** (3.1 mmol) and dicyclohexylurea (3.1 mmol) in 20 mL of toluene was refluxed for 2 h (until the disappearance of the dark violet color of compound **2**). Then, the resulting white precipitate was filtered off to afford the desired compound **1**.

*9-Benzoyl-1,3-dicyclohexyl-8-hydroxy-6-(2-hydroxyphenyl)-1,3,6-triazaspiro[4.4]non-8-ene-2,4,7-trione***(1a).** Yield: 1.58 g (94%); white solid; mp 285–287 °C; ^1^H NMR (400 MHz, DMSO-*d*_6_): δ = 9.90 (s, 1H), 7.75 (m, 2H), 7.62 (m, 1H), 7.51 (m, 2H), 7.24 (m, 1H), 7.00 (m, 1H), 6.91 (m, 1H), 6.81(m, 1H), 3.84 (m, 1H), 3.08 (m, 1H), 2.16–1.96 (m, 2H), 1.87–1.55 (m, 8H), 1.46 (m, 2H), 1.36–1.10 (m, 7H), 0.95 (m, 1H) ppm; ^13^C NMR (100 MHz, DMSO-*d*_6_): δ = 188.5, 169.2, 163.1, 155.5, 154.0, 153.9, 137.3, 132.8, 130.0, 128.6 (2C), 128.2 (2C), 126.7, 120.0, 119.0, 116.7, 113.1, 80.6, 52.1, 51.0, 30.0, 29.5, 28.6 (2C), 25.7, 25.2, 25.2, 25.0, 24.8 (2C) ppm. IR (mineral oil): 3354, 3151, 1779, 1723, 1708, 1678 cm^−1^. Anal. Calcd (%) for C_31_H_33_N_3_O_6_: C 68.49; H 6.12; N 7.73. Found: C 68.23; H 6.13; N 7.76. MS (ESI+): *m*/*z* calcd for C_31_H_33_N_3_O_6_+H^+^: 544.24 [M + H^+^]; found: 544.18.

*1,3-Dicyclohexyl-9-(4-ethoxybenzoyl)-8-hydroxy-6-(2-hydroxyphenyl)-1,3,6-triazaspiro[4.4]non-8-ene-2,4,7-trione***(1b).** Yield: 1.78 g (98%); white solid; mp 274–276 °C; ^1^H NMR (400 MHz, DMSO-*d*_6_): δ = 9.88 (s, 1H), 7.75 (m, 2H), 7.23 (m, 1H), 7.03–6.98 (m, 3H), 6.90 (m, 1H), 6.81 (m, 1H), 4.15 (q, *J* = 7.0 Hz, 2H), 3.84 (m, 1H), 3.05 (m, 1H), 2.15–1.97 (m, 2H),1.83–1.56 (m, 8H), 1.49–1.41 (m, 2H), 1.37–1.24 (m, 5H), 1.21–1.07 (m, 5H), 0.94 (m, 1H) ppm; ^13^C NMR (100 MHz, DMSO-*d*_6_): δ = 186.9, 169.2, 163.2, 162.5, 154.1, 153.9, 131.1 (2C), 130.0, 129.6, 128.1, 126.7, 120.1, 118.9, 116.6, 113.9 (2C), 111.9, 80.7, 63.5, 52.1, 51.0, 30.0, 29.5, 28.7 (2C), 28.6, 25.7, 25.2, 25.0, 24.8 (2C), 14.4 ppm. IR (mineral oil): 3386, 3173, 1777, 1727, 1714, 1683 cm^−1^. Anal. Calcd (%) for C_33_H_37_N_3_O_7_: C 67.45; H 6.35; N 7.15. Found: C 67.63; H 6.39; N 7.21. MS (ESI+): *m*/*z* calcd for C_33_H_37_N_3_O_7_+H^+^: 588.27 [M + H^+^]; found: 588.24.

*1,3-Dicyclohexyl-8-hydroxy-6-(2-hydroxyphenyl)-9-(4-methoxybenzoyl)-1,3,6-triazaspiro[4.4]non-8-ene-2,4,7-trione***(1c).** Yield: 1.60 g (90%); white solid; mp 274–276 °C; ^1^H NMR (400 MHz, DMSO-*d*_6_): δ = 9.88 (s, 1H), 7.77 (m, 2H), 7.24 (m, 1H), 7.05–6.98 (m, 3H), 6.91 (m, 1H), 6.81 (m, 1H), 3.86–3.81 (m, 4H), 3.06 (m, 1H), 2.15–1.97 (m, 2H), 1.83–1.70 (m, 5H), 1.66–1.54 (m, 3H), 1.49–1.41 (m, 2H), 1.35–1.25 (m, 2H), 1.21–1.08 (m, 5H), 0.94 (m, 1H) ppm; ^13^C NMR (100 MHz, DMSO-*d*_6_): δ = 186.9, 169.2, 163.2, 154.1, 153.9, 131.1 (2C), 130.0, 129.8, 128.8, 128.1, 126.7, 120.1, 118.9, 116.6, 113.8, 113.5 (2C), 80.7, 55.4, 52.1, 51.0, 30.0, 29.5, 28.7 (2C), 28.6, 25.7, 25.2, 25.0, 24.8 (2C) ppm. IR (mineral oil): 3385, 3159, 1778, 1727, 1716, 1683 cm^−1^. Anal. Calcd (%) for C_32_H_35_N_3_O_7_: C 67.00; H 6.15; N 7.33. Found: C 66.78; H 6.50; N 7.36. MS (ESI+): *m*/*z* calcd for C_32_H_35_N_3_O_7_+H^+^: 574.26 [M + H^+^]; found: 574.18.

*9-Benzoyl-6-(5-chloro-2-hydroxyphenyl)-1,3-dicyclohexyl-8-hydroxy-1,3,6-triazaspiro[4.4]non-8-ene-2,4,7-trione***(1d).** Yield: 1.56 g (87%); white solid; mp 294–296 °C; ^1^H NMR (400 MHz, DMSO-*d*_6_): δ = 9.88 (s, 1H), 7.74–7.68 (m, 4H), 7.24 (m, 1H), 6.99 (m, 1H), 6.90 (m, 1H), 6.81 (m, 1H), 3.84 (m, 1H), 3.09 (m, 1H), 2.12–1.98 (m, 2H), 1.84–1.58 (m, 8H), 1.47 (m, 2H), 1.36–1.05 (m, 7H), 0.95 (m, 1H) ppm; ^13^C NMR (100 MHz, DMSO-*d*_6_): δ = 188.5, 169.4, 163.3, 155.7, 153.9, 153.4, 137.3, 132.9, 129.9, 128.7 (2C), 128.3 (2C), 126.3, 121.9, 121.2, 118.2, 113.0, 80.6, 52.2, 51.1, 30.1, 29.5, 29.0, 28.7 (2C), 25.7, 25.3, 25.0, 24.9 (2C) ppm. IR (mineral oil): 3340, 3190, 1782, 1726, 1706, 1672 cm^−1^. Anal. Calcd (%) for C_31_H_32_ClN_3_O_6_: C 64.41; H 5.58; N 7.27. Found: C 64.67; H 5.72; N 7.24. MS (ESI+): *m*/*z* calcd for C_31_H_32_ClN_3_O_6_+H^+^: 578.21 [M + H^+^]; found: 578.15.

*1,3-Dicyclohexyl-8-hydroxy-6-(2-hydroxyphenyl)-9-(4-methylbenzoyl)-1,3,6-triazaspiro[4.4]non-8-ene-2,4,7-trione***(1e).** Yield: 1.52 g (88%); white solid; mp 291–293 °C; ^1^H NMR (400 MHz, DMSO-*d*_6_): δ = 9.90 (s, 1H), 7.66 (m, 2H), 7.32 (m, 2H), 7.24 (m, 1H), 7.00 (m, 1H), 6.90 (m, 1H), 6.81 (m, 1H), 3.84 (m, 1H), 3.07 (m, 1H), 2.39 (s, 3H), 2.15–1.96 (m, 2H), 1.83–1.54 (m, 8H), 1.47 (m, 2H), 1.36–1.10 (m, 7H), 0.94 (m, 1H) ppm; ^13^C NMR (100 MHz, DMSO-*d*_6_): δ = 188.1, 169.3, 163.2, 154.1, 153.9, 143.4, 134.7, 130.0 (2C), 128.8 (2C), 128.1, 126.7, 120.1, 119.0, 116.7, 113.4, 80.7, 52.1, 51.0, 30.0, 29.5, 28.7 (2C), 28.6, 25.7, 25.3, 25.2, 25.0, 24.8 (2C), 21.1 ppm. IR (mineral oil): 3382, 3182, 1779, 1727, 1715, 1682 cm^−1^. Anal. Calcd (%) for C_32_H_35_N_3_O_6_: C 68.92; H 6.33; N 7.54. Found: C 69.11; H 6.38; N 7.57. MS (ESI+): *m*/*z* calcd for C_32_H_35_N_3_O_6_+H^+^: 558.26 [M + H^+^]; found: 558.22.

*1,3-Dicyclohexyl-8-hydroxy-6-(2-hydroxyphenyl)-9-(4-nitrobenzoyl)-1,3,6-triazaspiro[4.4]non-8-ene-2,4,7-trione***(1f).** Yield: 1.70 g (93%); white solid; mp 279–281 °C; ^1^H NMR (400 MHz, DMSO-*d*_6_): δ = 9.91 (s, 1H), 8.31 (m, 2H), 7.96 (m, 2H), 7.24 (m, 1H), 7.00 (m, 1H), 6.91 (m, 1H), 6.81 (m, 1H), 3.84 (m, 1H), 3.12 (m, 1H), 2.16–1.95 (m, 2H), 1.89–1.59 (m, 8H), 1.49 (m, 2H), 1.36–1.10 (m, 7H), 0.95 (m, 1H) ppm; ^13^C NMR (100 MHz, DMSO-*d*_6_): δ = 186.5, 169.5, 163.2, 158.8, 153.9, 149.4, 142.9, 129.8 (2C), 128.8, 128.1, 126.7, 123.3 (2C), 120.2, 118.9, 116.6, 111.4, 80.4, 52.0, 51.0, 30.0, 29.5, 28.7 (2C), 25.7, 25.2, 25.2, 24.9 (2C), 24.9 ppm. IR (mineral oil): 3381, 3126, 1782, 1727, 1704, 1673 cm^−1^. Anal. Calcd (%) for C_31_H_32_N_4_O_8_: C 63.26; H 5.48; N 9.52. Found: C 63.29; H 5.50; N 9.53. MS (ESI+): *m*/*z* calcd for C_31_H_32_N_4_O_8_+H^+^: 589.23 [M + H^+^]; found: 589.19.

*9-(4-Chlorobenzoyl)-1,3-dicyclohexyl-8-hydroxy-6-(2-hydroxyphenyl)-1,3,6-triazaspiro[4.4]non-8-ene-2,4,7-trione***(1g).** Yield: 1.54 g (86%); white solid; mp 284–286 °C; ^1^H NMR (400 MHz, DMSO-*d*_6_): δ = 9.89 (s, 1H), 7.77 (m, 2H), 7.58 (m, 2H), 7.24 (m, 1H), 6.99 (m, 1H), 6.90 (m, 1H), 6.81 (m, 1H), 3.84 (m, 1H), 3.08 (m, 1H), 2.15–1.95 (m, 2H), 1.87–1.58 (m, 8H), 1.47 (m, 2H), 1.35–1.05 (m, 7H), 0.94 (m, 1H) ppm; ^13^C NMR (100 MHz, DMSO-*d*_6_): δ = 187.2, 169.2, 163.1, 156.3, 154.0, 153.8, 137.6, 136.1, 130.5 (2C), 130.0, 128.4 (2C), 126.7, 120.0, 118.9, 116.6, 112.6, 80.5, 52.0, 51.0, 30.0, 29.5, 28.7 (2C), 25.7, 25.2, 25.2, 24.9, 24.8 (2C) ppm. IR (mineral oil): 3369, 3135, 1781, 1729, 1709, 1658 cm^−1^. Anal. Calcd (%) for C_31_H_32_ClN_3_O_6_: C 64.41; H 5.58; N 7.27. Found: C 64.52; H 5.64; N 7.39. MS (ESI+): *m*/*z* calcd for C_31_H_32_ClN_3_O_6_+H^+^: 578.21 [M + H^+^]; found: 587.14.

#### 3.2.2. General Procedure to Compounds **4a–n**

A suspension of the corresponding compound **1** (1.8 mmol) and the corresponding carbodiimide **5** (1.8 mmol for **5a**,**c**,**d**; 3.6 mmol for DIC **5b**) in 20 mL of toluene was refluxed for 2 h. Then, the resulting precipitate was filtered off to afford the desired compound **4**.

*9-Benzoyl-1,3-dicyclohexyl-1-(1,3-dicyclohexyl-6-(2-hydroxyphenyl)-2,4,7-trioxo-1,3,6-triazaspiro[4.4]non-8-en-8-yl)urea***(4a).** Yield: 1.26 g (93%); white solid; mp 273–275 °C; ^1^H NMR (400 MHz, CDCl_3_): δ = 7.85 (m, 2H), 7.62 (m, 1H), 7.47 (m, 2H), 7.19 (m, 1H), 7.01 (m, 2H), 6.88 (m, 1H), 6.73 (br. s, 1H), 4.85 (d, 1H, *J* = 4.0 Hz), 4.03 (m, 1H), 3.74 (m, 1H), 3.46 (m, 1H), 2.99 (m, 1H), 2.21 (m, 2H), 2.00–1.50 (m, 22H), 1.44–0.97 (m, 16H) ppm; ^13^C NMR (100 MHz, CDCl_3_): δ = 189.4, 168.7, 165.8, 154.5, 154.4, 152.7, 142.5, 137.9, 135.8, 134.2, 130.2, 129.0 (2C), 128.7 (2C), 125.6, 121.7, 121.1, 119.4, 82.1, 59.7, 54.6, 52.8, 49.9, 33.3, 33.0, 31.9, 30.4, 30.4, 30.1, 29.1, 29.0, 26.3, 26.1, 26.1, 29.9, 25.8, 25.7, 25.6, 25.3, 25.2, 25.0, 24.9, 24.7 ppm. IR (mineral oil): 3411, 3132, 1781, 1730, 1652 cm^−1^. Anal. Calcd (%) for C_44_H_55_N_5_O_6_: C 70.47; H 7.39; N 9.34. Found: C 70.63; H 7.39; N 9.50. MS (ESI+): *m*/*z* calcd for C_44_H_55_N_5_O_6_+H^+^: 750.42 [M + H^+^]; found: 750.38.

*1-(9-Benzoyl-1,3-dicyclohexyl-6-(2-hydroxyphenyl)-2,4,7-trioxo-1,3,6-triazaspiro[4.4]non-8-en-8-yl)-1,3-diisopropylurea***(4b).** Yield: 1.06 g (88%); white solid; mp 231–232 °C; ^1^H NMR (400 MHz, DMSO-*d_6_*): δ = 9.96 (s, 1H), 7.85 (m, 2H), 7.68 (m, 1H), 7.52 (m, 2H), 7.23 (m, 1H), 6.99 (m, 1H) 6.91 (m, 1H), 6.82 (m, 1H), 5.78 (d, 1H, *J* = 4.0 Hz), 4.08 (m, 1H), 3.88 (m, 1H), 3.63 (m, 1H), 3.26 (s, 1H), 3.05 (m, 1H), 2.15–2.03 (m, 2H), 1.81–1.57 (m, 7H), 1.46 (m, 1H), 1.36–1.28 (m, 3H), 1.21–1.11 (m, 5H), 1.06–0.85 (m, 13H) ppm; ^13^C NMR (100 MHz, DMSO-*d_6_*): δ = 189.6, 168.4, 163.9, 154.2, 153.8, 153.6, 143.0, 136.6, 135.6, 134.1, 129.9, 128.6 (4C), 126.6, 120.5, 118.9, 116.6, 81.3, 52.9, 51.5, 49.7, 42.1, 29.8, 29.6, 28.7, 28.5, 25.6, 25.2, 25.1, 25.0, 24.7, 24.5, 22.4, 22.3, 21.4, 19.8 ppm. IR (mineral oil): 3404, 3171, 1771, 1747, 1711, 1671 cm^−1^. Anal. Calcd (%) for C_38_H_47_N_5_O_6_: C 68.14; H 7.07; N 10.46. Found: C 67.97; H 7.26; N 10.22. MS (ESI+): *m*/*z* calcd for C_38_H_47_N_5_O_6_+H^+^: 670.36 [M + H^+^]; found: 670.30.

*1,3-Dicyclohexyl-1-(1,3-dicyclohexyl-9-(4-ethoxybenzoyl)-6-(2-hydroxyphenyl)-2,4,7-trioxo-1,3,6-triazaspiro[4.4]non-8-en-8-yl)urea***(4c).** Yield: 1.30 g (91%); white solid; mp 192–194 °C; ^1^H NMR (400 MHz, DMSO-*d*_6_): δ = 9.99 (s, 1H), 7.86 (m, 2H), 7.22 (m, 1H), 6.98 (m, 3H), 6.88 (m, 1H), 6.80 (m, 1H) 5.91 (d, 1H, *J* = 4.0 Hz), 4.20–4.08 (m, 2H), 3.86 (m, 1H), 3.67 (m, 1H), 3.42 (s, 1H), 2.99 (m, 1H), 2.13–2.00 (m, 2H), 1.84–1.45 (m, 18H), 1.38–0.88 (m, 23H) ppm; ^13^C NMR (100 MHz, DMSO-*d*_6_): δ = 187.8, 168.3, 164.2, 163.4, 154.3 (2C), 153.6, 142.1, 139.2, 131.5 (2C), 129.9, 128.5, 126.7, 120.7, 118.9, 116.6, 114.3 (2C), 81.4, 63.7, 57.0, 53.0, 51.4, 49.2, 32.6 (2C), 31.3, 30.3, 29.9, 29.6, 28.7, 28.7, 25.6, 25.3, 25.2 (2C), 25.2, 24.9, 24.8, 24.7 (2C), 24.7 (2C), 24.6, 14.33 ppm. IR (mineral oil): 3357, 3175, 1775, 1731, 1716, 1646 cm^−1^. Anal. Calcd (%) for C_46_H_59_N_5_O_7_: C 69.58; H 7.49; N 8.82. Found: C 69.71; H 7.45; N 8.89. MS (ESI+): *m*/*z* calcd for C_46_H_59_N_5_O_7_+H^+^: 794.45 [M + H^+^]; found: 794.39.

*1,3-Dicyclohexyl-1-(1,3-dicyclohexyl-6-(2-hydroxyphenyl)-9-(4-methoxybenzoyl)-2,4,7-trioxo-1,3,6-triazaspiro[4.4]non-8-en-8-yl)urea***(4d).** Yield: 1.28 g (91%); white solid; mp 230–232 °C; ^1^H NMR (400 MHz, DMSO-*d*_6_): δ = 9.96 (s, 1H), 7.88 (m, 2H), 7.22 (m, 1H), 6.98 (m, 3H), 6.89 (m, 1H), 6.81 (m, 1H), 5.87 (d, 1H, *J* = 8.0 Hz), 3.91–3.84 (m, 4H), 3.67 (m, 1H), 3.43 (m, 1H), 2.99 (m, 1H), 2.11–2.01 (m, 2H), 1.85–1.45 (m, 18H), 1.39–0.86 (m, 20H) ppm; ^13^C NMR (100 MHz, DMSO-*d*_6_): δ = 187.8, 168.2, 164.1, 164.0, 154.2 (2C), 153.6, 142.1, 139.1, 131.4 (2C), 129.8, 128.6, 126.6, 120.7, 118.9, 116.5, 113.9 (2C), 81.36, 57.0, 55.6, 53.0, 51.4, 49.2, 32.5 (2C), 32.5, 31.2, 30.2, 29.8, 29.6, 28.7, 28.6, 25.6, 25.3, 25.2 (2C), 25.1, 24.8, 24.7, 24.7, 24.6 (2C), 24.5 ppm. IR (mineral oil): 3341, 3176, 1779, 1732, 1716, 1645 cm^−1^. Anal. Calcd (%) for C_45_H_57_N_5_O_7_: C 69.30; H 7.37; N 8.98. Found: C 69.53; H 7.45; N 8.89. MS (ESI+): *m*/*z* calcd for C_45_H_57_N_5_O_7_+H^+^: 780.43 [M + H^+^]; found: 780.37.

*1-(9-Benzoyl-6-(5-chloro-2-hydroxyphenyl)-1,3-dicyclohexyl-2,4,7-trioxo-1,3,6-triazaspiro[4.4]non-8-en-8-yl)-1,3-dicyclohexylurea***(4e).** Yield: 1.33 g (94%); white solid; mp 258–260 °C; ^1^H NMR (400 MHz, DMSO-*d*_6_): δ = 10.47 (s, 1H), 7.88 (m, 2H), 7.69 (m, 1H), 7.50 (m, 2H), 7.31 (m, 1H), 7.01 (m, 1H) 6.91 (m, 1H), 6.00 (d, 1H, *J* = 8.0 Hz), 3.89 (m, 1H), 3.63 (m, 1H), 3.53 (m, 1H), 3.01 (m, 1H), 2.15–2.02 (m, 2H), 1.84–1.45 (m, 18H), 1.37–0.83 (m, 20H) ppm; ^13^C NMR (100 MHz, DMSO-*d*_6_): δ = 189.7, 168.3, 164.1, 154.2, 153.6 (2C), 143.1, 137.7, 135.7, 134.5, 129.8, 128.9 (2C), 128.7 (2C), 126.4, 121.9, 121.7, 118.2, 81.3, 57.5, 53.1, 51.6, 49.4, 32.6 (2C), 32.5, 31.3, 30.3, 30.1, 29.7, 29.0, 28.7, 25.7, 25.4 (2C), 25.3 (2C), 25.2, 25.0, 24.9, 24.8 (2C), 24.6 ppm. IR (mineral oil): 3412, 3198, 1782, 1727, 1657 cm^−1^. Anal. Calcd (%) for C_44_H_54_ClN_5_O_6_: C 67.37; H 6.94; N 8.93. Found: C 67.43; H 6.88; N 9.02. MS (ESI+): *m*/*z* calcd for C_44_H_54_ClN_5_O_6_+H^+^: 783.38 [M + H^+^]; found: 783.32.

*1,3-Dicyclohexyl-1-(1,3-dicyclohexyl-6-(2-hydroxyphenyl)-9-(4-methylbenzoyl)-2,4,7-trioxo-1,3,6-triazaspiro[4.4]non-8-en-8-yl)urea***(4f).** Yield: 1.22 g (89%); white solid; mp 245–253 °C; ^1^H NMR (400 MHz, DMSO-*d*_6_): δ = 10.00 (s, 1H), 7.77 (m, 2H), 7.30 (m, 2H), 7.22 (m, 1H), 6.98 (m, 1H), 6.89 (m, 1H) 6.81 (m, 1H), 5.84 (d, 1H, *J* = 8.0 Hz), 3.87 (m, 1H), 3.67 (m, 1H), 3.38 (m, 1H), 3.00 (m, 1H), 2.39 (s, 3H), 2.14–2.00 (m, 2H), 1.84–1.45 (m, 18H), 1.41–0.85 (m, 20H) ppm; ^13^C NMR (100 MHz, DMSO-*d*_6_): δ = 189.2, 168.3, 164.0, 154.2, 154.1, 153.6, 145.0, 142.7, 138.3, 133.3 (2C), 129.9, 129.2 (2C), 128.9, 126.7, 120.6, 119.0, 116.6, 81.4, 59.6, 57.2, 53.0, 51.4, 49.2, 32.6, 32.5, 31.2, 30.2, 29.9, 29.6, 28.7, 28.6, 25.6, 25.4, 25.2 (2C), 25.2, 25.1, 24.8, 24.8, 24.7, 24.6 (2C), 21.3 ppm. IR (mineral oil): 3336, 3164, 1778, 1734, 1715, 1650 cm^−1^. Anal. Calcd (%) for C_45_H_57_N_5_O_6_: C 70.75; H 7.52; N 9.17. Found: C 70.91; H 7.62; N 9.24. MS (ESI+): *m*/*z* calcd for C_45_H_57_N_5_O_6_+H^+^: 764.44 [M + H^+^]; found: 764.34.

*1,3-Dicyclohexyl-1-(1,3-dicyclohexyl-6-(2-hydroxyphenyl)-9-(4-nitrobenzoyl)-2,4,7-trioxo-1,3,6-triazaspiro[4.4]non-8-en-8-yl)urea***(4g).** Yield: 1.27 g (89%); white solid; mp 229–231 °C; ^1^H NMR (400 MHz, DMSO-*d*_6_): δ = 10.10 (s, 1H), 8.33 (m, 2H), 8.01 (m, 2H), 7.24 (m, 1H), 6.99 (m, 1H), 6.90 (m, 1H) 6.83 (m, 1H), 5.99 (d, 1H, *J* = 8.0 Hz), 3.87 (m, 1H), 3.73 (m, 1H), 3.49 (m, 1H), 3.06 (m, 1H), 2.14–1.97 (m, 2H), 1.81–1.41 (m, 22H), 1.36–0.88 (m, 16H) ppm; ^13^C NMR (100 MHz, DMSO-*d*_6_): δ = 188.6, 168.4, 163.8, 154.2, 154.1, 153.6, 150.2, 144.9, 140.6, 138.3, 130.2, 129.9 (2C), 126.8, 123.9 (2C), 120.4, 119.1, 116.7, 81.1, 57.8, 52.9, 51.6, 49.5, 32.5, 32.3 (2C), 29.9, 29.8, 28.7, 28.7, 25.7, 25.6, 25.4, 25.3, 25.2 (2C), 25.0, 24.9 (2C), 24.7, 24.7 (2C), 24.6 ppm. IR (mineral oil): 3336, 3194, 1779, 1731, 1716, 1656 cm^−1^. Anal. Calcd (%) for C_44_H_54_N_6_O_8_: C 66.48; H 6.85; N 10.57. Found: C 66.35; H 6.89; N 10.44. MS (ESI+): *m*/*z* calcd for C_44_H_54_N_6_O_8_+H^+^: 794.40 [M + H^+^]; found: 794.36.

*1-(9-(4-Chlorobenzoyl)-1,3-dicyclohexyl-6-(2-hydroxyphenyl)-2,4,7-trioxo-1,3,6-triazaspiro[4.4]non-8-en-8-yl)-1,3-dicyclohexylurea***(4h).** Yield: 1.31 g (93%); white solid; mp 203–204 °C; ^1^H NMR (400 MHz, DMSO-*d*_6_): δ = 9.98 (s, 1H), 7.86 (m, 2H), 7.58 (m, 2H), 7.23 (m, 1H), 6.99 (m, 1H), 6.90 (m, 1H), 6.81 (m, 1H), 5.94 (d, 1H, *J* = 4.0 Hz), 3.87 (m, 1H), 3.73 (m, 1H), 3.33 (m, 1H), 3.03 (m, 1H), 2.14–1.99 (m, 2H), 1.82–1.40 (m, 20H), 1.34–0.87 (m, 18H) ppm; ^13^C NMR (100 MHz, DMSO-*d*_6_): δ = 188.5, 168.3, 163.8, 154.2, 154.0, 153.5, 143.7, 139.0, 134.4, 130.4, 129.9, 128.8 (2C), 126.7, 120.5 (2C), 118.9, 116.6, 81.1, 57.4, 52.9, 51.4, 49.3, 32.5, 32.3 (2C), 31.2, 30.0, 29.8, 29.6, 28.6, 28.6 (2C), 25.6, 25.4, 25.2 (2C), 25.2, 25.1, 25.0, 24.8, 24.6 (2C), 24.6 ppm. IR (mineral oil): 3345, 3181, 1780, 1731, 1718, 1655 cm^−1^. Anal. Calcd (%) for C_44_H_54_ClN_5_O_6_: C 67.37; H 6.94; N 8.93. Found: C 67.76; H 7.11; N 9.02. MS (ESI+): *m*/*z* calcd for C_44_H_54_ClN_5_O_6_+H^+^: 784.38 [M + H^+^]; found: 784.33. Crystal structure of compound **4h** was deposited at the Cambridge Crystallographic Data Centre with the deposition number CCDC 2090986.

*1-(1,3-Dicyclohexyl-6-(2-hydroxyphenyl)-9-(4-methylbenzoyl)-2,4,7-trioxo-1,3,6-triazaspiro[4.4]non-8-en-8-yl)-1,3-diisopropylurea***(4i).** Yield: 1.05 g (85%); yellow solid; mp 158–160 °C; ^1^H NMR (400 MHz, DMSO-*d*_6_): δ = 9.96 (s, 1H), 7.76 (m, 2H), 7.32 (m, 2H), 7.23 (m, 1H), 6.99 (m, 1H) 6.90 (m, 1H), 6.82 (m, 1H), 5.73 (d, 1H, *J* = 8.0 Hz), 4.08 (m, 1H), 3.87 (m, 1H), 3.64 (m, 1H), 3.26 (s, 1H), 3.03 (m, 1H), 2.39 (s, 3H), 2.14–2.03 (m, 2H), 1.85–1.53 (m, 7H), 1.47 (m, 1H), 1.37–1.28 (m, 3H), 1.20–1.11 (m, 5H), 1.04–0.86 (m, 13H) ppm; ^13^C NMR (100 MHz, DMSO-*d*_6_): δ = 189.1, 168.4, 164.0, 154.1, 153.8, 153.6, 144.8, 142.5, 139.7, 133.1, 130.0, 129.2 (2C), 128.8 (2C), 126.5, 120.6, 118.9, 116.5, 81.3, 52.9, 51.5, 49.6, 42.1, 29.8, 29.6, 28.7, 28.5, 25.6, 25.2, 25.1, 25.0, 24.7, 24.5, 22.4, 22.3, 21.4, 21.17, 19.8 ppm. IR (mineral oil): 3324, 3180, 1799, 1748, 1693, 1669, 1644 cm^−1^. Anal. Calcd (%) for C_39_H_49_N_5_O_6_: C 68.50; H 7.22; N 10.24. Found: C 68.65; H 7.26; N 10.30. MS (ESI+): *m*/*z* calcd for C_39_H_49_N_5_O_6_+H^+^: 684.38 [M + H^+^]; found: 684.34.

*1-(1,3-Dicyclohexyl-6-(2-hydroxyphenyl)-9-(4-methoxybenzoyl)-2,4,7-trioxo-1,3,6-triazaspiro[4.4]non-8-en-8-yl)-1,3-diisopropylurea***(4j).** Yield: 1.08 g (86%); white solid; mp 150–152 °C; ^1^H NMR (400 MHz, CDCl_3_): δ = 7.86 (m, 2H), 7.18 (m, 1H), 7.01 (m, 3H), 6.94 (m, 2H), 6.86 (m, 1H), 4.88 (d, 1H, *J* = 8.0 Hz), 4.22 (m, 1H), 4.02 (m, 1H), 3.88 (s, 3H), 3.76 (m, 1H), 2.97 (m, 1H), 2.27–2.14 (m, 2H), 2.01 (m, 1H), 1.92–1.69 (m, 6H), 1.60 (m, 1H), 1.51–1.47 (m, 2H), 1.44–0.88 (m, 20H) ppm; ^13^C NMR (100 MHz, CDCl_3_): δ = 187.5, 168.8, 165.8, 164.7, 154.5, 154.4, 152.8, 141.5, 138.6, 131.6 (2C), 130.2, 128.5, 125.8, 121.6, 120.9, 119.1, 114.1 (2C), 82.3, 55.5, 54.6, 52.8, 51.2, 43.1, 30.4, 30.2, 29.2, 28.9, 26.1, 25.9, 25.9 (2C), 25.2, 25.0, 23.0, 22.6, 22.0, 20.2 ppm. IR (mineral oil): 3421, 3198, 1716, 1686, 1664, 1649 cm^−1^. Anal. Calcd (%) for C_39_H_49_N_5_O_7_: C 66.93; H 7.06; N 10.01. Found: C 67.02; H 7.00; N 10.05. MS (ESI+): *m*/*z* calcd for C_39_H_49_N_5_O_7_+H^+^: 700.37 [M + H^+^]; found: 700.32.

*1-(9-(4-Chlorobenzoyl)-1,3-dicyclohexyl-6-(2-hydroxyphenyl)-2,4,7-trioxo-1,3,6-triazaspiro[4.4]non-8-en-8-yl)-1,3-diisopropylurea***(4k).** Yield: 1.12 g (88%); white solid; mp 154–157 °C; ^1^H NMR (400 MHz, CDCl_3_): δ = 7.80 (m, 2H), 7.46 (m, 2H), 7.19 (m, 1H), 6.99 (m, 2H), 6.87 (m, 1H), 4.87 (d, 1H, *J* = 4.0 Hz), 4.20 (m, 1H), 4.02 (m, 1H), 3.72 (m, 1H), 2.96 (m, 1H), 2.20 (m, 2H), 2.00 (m, 1H), 1.92–1.70 (m, 6H), 1.63–1.62 (m, 1H), 1.51–1.49 (m, 2H), 1.44–1.23 (m, 6H), 1.18–1.00 (m, 14H) ppm; ^13^C NMR (100 MHz, CDCl_3_): δ = 188.1, 168.7, 165.4, 154.5, 154.3, 152.8, 142.7, 141.0, 137.2, 134.0, 130.3 (3C), 129.2 (2C), 125.7, 121.3, 121.0, 119.0, 82.1, 54.6, 52.8, 51.3, 43.2, 30.4, 30.3, 29.1, 28.9, 26.1, 25.9, 25.8, 25.2, 25.0, 23.4, 23.0, 22.6, 21.9, 20.2 ppm. IR (mineral oil): 3173, 3083, 1702, 1681, 1666 cm^−1^. Anal. Calcd (%) for C_38_H_46_ClN_5_O_6_: C 68.81; H 6.58; N 9.94. Found: C 68.85; H 6.63; N 9.89. MS (ESI+): *m*/*z* calcd for C_38_H_46_ClN_5_O_6_+H^+^: 704.32 [M + H^+^]; found: 704.30.

*9-Benzoyl-1-(1,3-dicyclohexyl-6-(2-hydroxyphenyl)-2,4,7-trioxo-1,3,6-triazaspiro[4.4]non-8-en-8-yl)-1,3-diphenylurea***(4l).** Yield: 1.14 g (86%); pale yellow solid; mp 148–150 °C; ^1^H NMR (400 MHz, CDCl_3_): δ = 7.69 (m, 2H), 7.55 (m, 1H), 7.39 (m, 2H), 7.27–7.17 (m, 4H), 7.16–7.03 (m, 7H), 6.88 (m, 2H), 6.80–6.67 (m, 3H), 3.97 (m, 1H), 3.12 (m, 1H), 2.14 (m, 2H), 2.96–0.95 (m, 18H) ppm; ^13^C NMR (100 MHz, CDCl_3_): δ = 188.1, 168.1, 164.5, 154.4, 152.7, 152.3, 144.2, 140.4, 137.1, 136.6, 134.4, 134.1, 130.3, 129.9 (2C), 129.0 (2C), 128.9, 128.7 (2C), 127.0, 125.8, 125.1 (2C), 124.5, 121.9, 121.8, 121.2, 120.0 (2C), 119.8, 82.0, 54.4, 52.7, 30.5, 30.2, 29.0 (2C), 26.1, 26.0, 25.8, 25.7, 25.1, 25.1 ppm. IR (mineral oil): 3169, 1778, 1721, 1647 cm^−1^. Anal. Calcd (%) for C_44_H_43_N_5_O_6_: C 71.62; H 5.87; N 9.49. Found: C 71.89; H 5.91; N 9.55. MS (ESI+): *m*/*z* calcd for C_44_H_43_N_5_O_6_+H^+^: 738.33 [M + H^+^]; found: 738.26.

*1-(1,3-Dicyclohexyl-6-(2-hydroxyphenyl)-2,4,7-trioxo-1,3,6-triazaspiro[4.4]non-8-en-8-yl)-9-(4-ethoxybenzoyl)-1,3-diphenylurea***(4m).** Yield: 1.21 g (86%); yellow solid; mp 161–163 °C; ^1^H NMR (400 MHz, CDCl_3_): δ = 7.72 (m, 2H), 7.32–7.05 (m, 11H), 6.94–6.85 (m, 3H), 6.78 (m, 2H), 6.71 (m, 1H), 4.09 (q, *J* = 6.8 Hz, 2H), 3.98 (m, 1H), 3.10 (m, 1H), 2.16 (m, 2H), 1.91–0.99 (m, 21H) ppm; ^13^C NMR (100 MHz, CDCl_3_): δ = 186.2, 168.1, 164.6, 164.1, 154.4, 152.6, 152.3, 143.4, 140.4, 137.2, 135.8, 131.4 (2C), 130.2, 129.8 (2C), 129.3, 129.0 (2C), 127.1, 125.7, 125.3 (2C), 124.4, 122.0, 121.3, 120.0 (3C), 114.5 (2C), 82.0, 64.0, 54.5, 52.7, 30.5, 30.2, 29.0 (2C), 26.1, 25.9, 25.8 (2C), 25.1 (2C), 14.5 ppm. IR (mineral oil): 3327, 3183, 1778, 1721, 1644 cm^−1^. Anal. Calcd (%) for C_46_H_47_N_5_O_7_: C 70.66; H 6.06; N 8.96. Found: C 70.34; H 6.14; N 9.05. MS (ESI+): *m*/*z* calcd for C_46_H_47_N_5_O_7_+H^+^: 782.36 [M + H^+^]; found: 782.36.

*1,3-Dibutyl-1-(1,3-dicyclohexyl-6-(2-hydroxyphenyl)-2,4,7-trioxo-1,3,6-triazaspiro[4.4]non-8-en-8-yl)-9-(4-ethoxybenzoyl)urea***(4n).** To precipitate compound **4n**, hexane (50 mL) was added to the reaction mixture. Yield: 1.19 g (89%); pale yellow solid; mp 138–140 °C; ^1^H NMR (400 MHz, CDCl_3_): δ = 7.71 (m, 2H), 7.20 (m, 1H), 7.03–6.85 (m, 5H), 5.05 (m, 1H), 4.11 (q, *J* = 6.8 Hz, 2H), 4.05–3.97 (m, 1H), 3.71–3.64 (m, 1H), 3.51–3.44 (m, 1H), 3.05–2.97 (m, 1H), 2.95–2.84 (m, 2H), 2.19 (m, 2H), 2.05–0.87 (m, 35H) ppm; ^13^C NMR (100 MHz, CDCl_3_): δ = 187.0, 169.0, 165.4, 164.1, 155.1, 154.4, 152.6, 142.6, 132.4, 131.0 (2C), 130.3, 128.0, 125.5, 121.8, 121.2, 119.5, 114.7 (2C), 82.1, 63.9, 54.6, 52.8, 48.7, 40.7, 31.4 (2C), 31.3, 30.4, 30.1, 29.1, 28.9, 26.1, 25.9, 25.8, 25.2, 25.0, 20.1, 20.0, 14.6, 13.8 (2C) ppm. IR (mineral oil): 3538, 3271, 1780, 1724, 1642 cm^−1^. Anal. Calcd (%) for C_42_H_55_N_5_O_7_: C 67.99; H 7.47; N 9.44. Found: C 67.64; H 7.53; N 9.31. MS (ESI+): *m*/*z* calcd for C_42_H_55_N_5_O_7_+H^+^: 742.42 [M + H^+^]; found: 742.37.

#### 3.2.3. General Procedure to Compounds **3a–n**

The corresponding compound **4** (0.5 mmol) was put into an oven-dried tube, pressed slightly, and then, it was heated at 160–290 °C (the temperature for each compound is given in Table 2; caution: R′NCO evolves during the reaction). The reaction mixture was cooled to room temperature and scrubbed with hexane (about 10 mL) to give the appropriate compound **3**.

*9-Benzoyl-1,3-dicyclohexyl-8-(cyclohexylamino)-6-(2-hydroxyphenyl)-1,3,6-triazaspiro[4.4]non-8-ene-2,4,7-trione***(3a).** Yield: 300 mg (96%); white solid; mp 215–217 °C; ^1^H NMR (400 MHz, CDCl_3_): δ = 7.69 (m, 2H), 7.59 (m, 1H), 7.51 (m, 2H), 7.24 (m, 1H), 7.08 (m, 1H), 7.03 (m, 1H), 6.91 (m, 1H), 6.45 (br. s, 1H), 5.60 (d, 1H, *J* = 8.0 Hz), 3.99 (m, 1H), 2.84 (m, 1H), 2.22–2.10 (m, 2H), 1.97–1.82 (m, 6H), 1.68–1.53 (m, 7H), 1.45–0.75 (m, 16H) ppm; ^13^C NMR (100 MHz, CDCl_3_): δ = 189.7, 170.3, 165.7, 154.7, 152.4, 139.9, 132.7, 130.4, 129.0 (2C), 128.1, 126.1, 122.1, 121.5 (2C), 120.0, 107.1, 83.1, 53.8, 52.3, 33.9, 32.9, 32.4, 30.5, 30.0, 29.1, 29.0, 26.4, 26.0, 25.9, 25.9, 25.2, 25.2, 25.1, 24.5, 24.2 ppm. IR (mineral oil): 3363, 3276, 1767, 1724, 1700 cm^−1^. Anal. Calcd (%) for C_37_H_44_N_4_O_5_: C 71.13; H 7.10; N 8.97. Found: C 71.20; H 7.17; N 8.93. MS (ESI+): *m*/*z* calcd for C_37_H_44_N_4_O_5_+H^+^: 625.34 [M + H^+^]; found: 625.31.

*9-Benzoyl-1,3-dicyclohexyl-6-(2-hydroxyphenyl)-8-(isopropylamino)-1,3,6-triazaspiro[4.4]non-8-ene-2,4,7-trione***(3b).** Yield: 263 mg (90%); white solid; mp 147–149 °C; ^1^H NMR (400 MHz, CDCl_3_): δ = 7.70 (m, 2H), 7.58 (m, 1H), 7.49 (m, 2H), 7.23 (m, 1H), 7.05 (m, 2H), 6.90 (m, 1H), 6.49 (m, 1H), 5.49 (d, 1H, *J* = 8.0 Hz), 3.98 (m, 1H), 3.29 (m, 1H), 2.84 (m, 1H), 2.22–2.09 (m, 2H), 1.97–1.76 (m, 5H), 1.74–1.66 (m, 3H), 1.58–1.51 (m, 2H), 1.39–1.04 (m, 8H), 0.99–0.94 (m, 6H) ppm; ^13^C NMR (100 MHz, CDCl_3_): δ = 189.6, 170.2, 165.6, 154.6, 152.4, 139.6, 132.8, 130.4, 129.0 (2C), 128.0 (2C), 126.0, 122.0, 121.5, 119.9, 107.7, 83.0, 53.8, 52.3, 46.9, 30.5, 30.0, 29.0, 29.0, 26.4, 26.0, 25.9, 25.8, 25.2, 25.2, 22.7, 22.4 ppm. IR (mineral oil): 3354, 3262, 3170, 1726 cm^−1^. Anal. Calcd (%) for C_34_H_40_N_4_O_5_: C 69.84; H 6.90; N 9.58. Found: C 69.93; H 6.84; N 9.61. MS (ESI+): *m*/*z* calcd for C_34_H_40_N_4_O_5_+H^+^: 585.31 [M + H^+^]; found: 585.30. Crystal structure of compound **3b** was deposited at the Cambridge Crystallographic Data Centre with the deposition number CCDC 2090985.

*1,3-Dicyclohexyl-8-(cyclohexylamino)-9-(4-ethoxybenzoyl)-6-(2-hydroxyphenyl)-1,3,6-triazaspiro[4.4]non-8-ene-2,4,7-trione***(3c).** Yield: 311 mg (93%); white solid; mp 152–154 °C; ^1^H NMR (400 MHz, CDCl_3_): δ = 7.70 (m, 2H), 7.23 (m, 1H), 7.07 (m, 1H), 7.02 (m, 1H), 6.97 (m, 2H), 6.90 (m, 1H), 6.45 (s, 1H), 5.46 (br. s, 1H), 4.17–4.09 (m, 2H), 4.00 (m, 1H), 2.81 (m, 1H), 2.24–2.12 (m, 2H), 1.93–1.80 (m, 7H), 1.68–1.44 (m, 11H), 1.38–0.76 (m, 14H) ppm; ^13^C NMR (100 MHz, CDCl_3_): δ = 188.6, 170.4, 166.0, 163.0, 154.7, 141.6, 132.2, 130.5 (2C), 130.3, 125.9, 122.2, 121.5 (2C), 120.0, 114.7, 107.8, 83.2, 63.9, 53.8, 52.3, 33.1, 32.4, 30.5, 30.0, 29.1, 29.0, 26.4, 26.0, 25.9, 25.9, 25.2, 25.2, 25.2 (2C), 24.6, 24.3, 14.6 ppm. IR (mineral oil): 3196, 1760, 1718, 1687, 1662 cm^−1^. Anal. Calcd (%) for C_39_H_48_N_4_O_6_: C 70.04; H 7.23; N 8.38. Found: C 70.16; H 7.20; N 8.44. MS (ESI+): *m*/*z* calcd for C_39_H_48_N_4_O_6_+H^+^: 669.37 [M + H^+^]; found: 669.32. Crystal structure of compounds **3c** was deposited at the Cambridge Crystallographic Data Centre with the deposition number CCDC 2090984.

*1,3-Dicyclohexyl-8-(cyclohexylamino)-6-(2-hydroxyphenyl)-9-(4-methoxybenzoyl)-1,3,6-triazaspiro[4.4]non-8-ene-2,4,7-trione***(3d).** Yield: 291 mg (91%); white solid; mp 275–277 °C; ^1^H NMR (400 MHz, CDCl_3_): δ = 7.71 (m, 2H), 7.23 (m, 1H), 7.08–6.97 (m, 4H), 6.90 (m, 1H), 6.46 (br. s, 1H), 5.48 (d, 1H, *J* = 12.0 Hz), 4.00 (m, 1H), 3.89 (s, 3H), 2.81 (m, 1H), 2.18 (m, 2H), 1.91–1.82 (m, 7H), 1.68–1.43 (m, 7H), 1.37–0.75 (m, 15H) ppm; ^13^C NMR (100 MHz, CDCl_3_): δ = 188.5, 170.4, 166.0, 163.6, 154.7, 152.3, 132.4, 130.5 (2C), 130.3, 125.9, 122.2, 121.5, 120.0, 114.2 (2C), 107.7, 83.0, 55.6, 53.8, 53.7, 52.3, 33.1, 32.4, 30.5, 30.0, 29.1, 29.0, 26.4, 26.0, 25.9, 25.9, 25.2, 25.2, 25.1, 24.6, 24.3 ppm. IR (mineral oil): 3376, 3282, 1764, 1723, 1700 cm^−1^. Anal. Calcd (%) for C_38_H_46_N_4_O_5_: C 69.70; H 7.08; N 8.56. Found: C 69.79; H 7.03; N 8.61. MS (ESI+): *m*/*z* calcd for C_38_H_46_N_4_O_5_+H^+^: 639.35 [M + H^+^]; found: 639.30.

*9-Benzoyl-6-(5-chloro-2-hydroxyphenyl)-1,3-dicyclohexyl-8-(cyclohexylamino)-1,3,6-triazaspiro[4.4]non-8-ene-2,4,7-trione***(3e).** Yield: 297 mg (90%); white solid; mp 240–242 °C; ^1^H NMR (400 MHz, DMSO-*d_6_*): δ = 10.23 (s, 1H), 7.66 (m, 3H), 7.57 (m, 2H), 7.30 (m, 1H), 7.00 (m, 1H), 6.90 (m, 1H), 3.85 (m, 1H), 3.01 (m, 1H), 2.16–2.02 (m, 2H), 1.84–1.72 (m, 5H), 1.66–1.13 (m, 17H), 1.07–0.83 (m, 6H), 0.69 (m, 1H), 0.35 (m, 1H) ppm; ^13^C NMR (100 MHz, DMSO-*d_6_*): δ = 189.6, 170.1, 163.5, 153.9, 153.2, 142.9, 139.8, 132.7, 129.7, 128.9 (2C), 127.9, 125.8, 121.8 (2C), 121.5, 118.2, 104.4, 82.3, 54.4, 51.9, 50.9, 31.4, 30.8, 30.2, 29.4, 28.9, 28.7, 25.7, 25.2, 25.1, 24.9, 24.8 (2C), 24.4, 24.3, 24.2 ppm. IR (mineral oil): 3362, 3280, 1767, 1723, 1700 cm^−1^. Anal. Calcd (%) for C_37_H_43_ClN_4_O_5_: C 67.41; H 6.58; N 8.50. Found: C 67.53; H 6.62; N 8.54. MS (ESI+): *m*/*z* calcd for C_37_H_43_ClN_4_O_5_+H^+^: 659.30 [M + H^+^]; found: 659.31.

*1,3-Dicyclohexyl-8-(cyclohexylamino)-6-(2-hydroxyphenyl)-9-(4-methylbenzoyl)-1,3,6-triazaspiro[4.4]non-8-ene-2,4,7-trione***(3f).** Yield: 313 mg (98%); white solid; mp 159–190 °C; ^1^H NMR (400 MHz, DMSO-*d_6_*): δ = 9.82 (s, 1H), 7.58 (m, 2H), 7.38 (m, 2H), 7.22 (m, 1H), 6.98 (m, 1H), 6.88 (m, 1H), 6.80 (m, 1H), 6.76 (m, 1H), 3.83 (m, 1H), 3.01 (m, 1H), 2.39 (s, 3H), 2.15–1.97 (m, 2H), 1.83–1.72 (m, 5H), 1.66–1.41 (m, 7H), 1.35–0.99 (m, 13H), 0.96–0.84 (m, 2H), 0.68 (m, 1H), 0.39 (m, 1H) ppm; ^13^C NMR (100 MHz, DMSO-*d_6_*): δ = 189.4, 170.2, 163.6, 154.0 (2C), 143.0 (2C), 137.4, 129.8, 129.3 (2C), 127.9 (2C), 126.2, 120.5, 118.8, 116.6, 104.6, 82.4, 54.1, 51.9, 50.8, 31.4, 31.1, 30.0, 29.4, 28.7 (2C), 25.7, 25.3, 25.2, 25.1, 24.8 (2C), 24.5, 24.3, 24.2, 21.0 ppm. IR (mineral oil): 3354, 3277, 1787, 1724, 1700 cm^−1^. Anal. Calcd (%) for C_38_H_46_N_4_O_5_: C 71.45; H 7.26; N 8.77. Found: C 71.37; H 7.23; N 8.85. MS (ESI+): *m*/*z* calcd for C_38_H_46_N_4_O_5_+H^+^: 639.35 [M + H^+^]; found: 639.30.

*1,3-Dicyclohexyl-8-(cyclohexylamino)-6-(2-hydroxyphenyl)-9-(4-nitrobenzoyl)-1,3,6-triazaspiro[4.4]non-8-ene-2,4,7-trione***(3g).** Yield: 301 mg (90%); white solid; mp 164–166 °C; ^1^H NMR (400 MHz, DMSO-*d_6_*): δ = 9.89 (s, 1H), 8.40 (m, 2H), 7.86 (m, 2H), 7.32 (m, 1H), 7.24 (m, 1H), 6.99 (m, 1H), 6.88 (m, 1H), 6.81 (m, 1H), 3.83 (m, 1H), 3.04 (m, 1H), 2.16–1.95 (m, 2H), 1.81–1.70 (m, 5H), 1.65–1.09 (m, 19H), 1.01–0.74 (m, 4H), 0.34 (br. s, 1H) ppm; ^13^C NMR (100 MHz, DMSO-*d_6_*): δ = 187.5, 170.1, 163.0, 154.0, 153.9, 149.5, 144.7, 144.3, 130.0, 129.4 (2C), 126.2, 124.2 (2C), 120.3, 118.9, 116.7, 82.2, 55.6, 51.84, 50.89, 31.53, 30.1, 29.5, 28.7 (2C), 25.8, 25.3, 25.2, 24.9, 24.9, 24.8, 24.4 (2C), 24.4, 24.3 ppm. IR (mineral oil): 3313, 3270, 1782, 1727, 1700 cm^−1^. Anal. Calcd (%) for C_37_H_43_N_5_O_7_: C 66.35; H 6.47; N 10.46. Found: C 66.44; H 6.53; N 10.58. MS (ESI+): *m*/*z* calcd for C_37_H_43_N_5_O_7_+H^+^: 670.32 [M + H^+^]; found: 670.25.

*9-(4-Chlorobenzoyl)-1,3-dicyclohexyl-8-(cyclohexylamino)-6-(2-hydroxyphenyl)-1,3,6-triazaspiro[4.4]non-8-ene-2,4,7-trione***(3h).** Yield: 297 mg (90%); white solid; mp 173–175 °C; ^1^H NMR (400 MHz, CDCl_3_): δ = 7.65 (m, 2H), 7.49 (m, 2H), 7.24 (m, 1H), 7.04 (m, 2H), 6.92 (m, 1H), 6.38 (s, 1H), 5.77 (br. s, 1H), 3.97 (m, 1H), 2.81 (m, 1H), 2.21–2.09 (m, 2H), 1.95–1.80 (m, 7H), 1.68–1.46 (m, 9H), 1.36–0.85 (m, 13H) ppm; ^13^C NMR (100 MHz, CDCl_3_): δ = 188.4, 170.1, 165.5, 154.6, 152.3, 139.2, 138.1, 130.5, 129.5 (2C), 129.3 (2C), 126.0, 122.0, 121.6, 120.0, 106.8, 83.1, 53.8, 52.3, 32.9, 32.4, 32.4, 30.5, 30.1, 29.1, 29.0, 26.4, 26.0, 25.9, 25.9, 25.2, 25.2, 25.1, 24.5, 24.3 ppm. IR (mineral oil): 3379, 3278, 1767, 1724, 1701, 1633 cm^−1^. Anal. Calcd (%) for C_37_H_43_ClN_4_O_5_: C 67.41; H 6.58; N 8.50. Found: C 67.53; H 6.61; N 8.47. MS (ESI+): *m*/*z* calcd for C_37_H_43_ClN_4_O_5_+H^+^: 659.30 [M + H^+^]; found: 659.24.

*1,3-Dicyclohexyl-6-(2-hydroxyphenyl)-8-(isopropylamino)-9-(4-methylbenzoyl)-1,3,6-triazaspiro[4.4]non-8-ene-2,4,7-trione***(3i).** Yield: 281 mg (94%); white solid; mp 146–148 °C; ^1^H NMR (400 MHz, CDCl_3_): δ = 7.62 (m, 2H), 7.28 (m, 1H), 7.21 (m, 1H), 7.07 (m, 1H), 7.00 (m, 1H), 6.89 (m, 1H), 6.58 (br. s, 1H), 5.43 (d, 1H, *J* = 12.0 Hz), 3.98 (m, 1H), 3.30 (m, 1H), 2.83 (m, 1H), 2.43 (s, 3H), 2.22–2.10 (m, 2H), 1.95–1.65 (m, 8H), 1.56–1.50 (m, 2H), 1.40–0.88 (m, 15H) ppm; ^13^C NMR (100 MHz, CDCl_3_): δ = 189.4, 170.2, 165.8, 154.6, 152.4, 143.7, 136.9, 130.3, 129.6 (2C), 128.3 (2C), 126.0, 122.0, 121.4, 119.8, 108.0, 83.1, 52.3, 46.8, 30.5, 30.0, 29.0 (2C), 26.4, 26.0, 25.9 (2C), 25.9, 25.2, 25.2, 22.7, 22.3, 21.6 ppm. IR (mineral oil): 3197, 1760, 1718, 1687 cm^−1^. Anal. Calcd (%) for C_35_H_42_N_4_O_5_: C 70.21; H 7.07; N 9.36. Found: C 70.45; H 7.10; N 9.30. MS (ESI+): *m*/*z* calcd for C_35_H_42_N_4_O_5_+H^+^: 599.32 [M + H^+^]; found: 599.32.

*1,3-Dicyclohexyl-6-(2-hydroxyphenyl)-8-(isopropylamino)-9-(4-methoxybenzoyl)-1,3,6-triazaspiro[4.4]non-8-ene-2,4,7-trione***(3j).** Yield: 280 mg (91%); white solid; mp 148–150 °C; ^1^H NMR (400 MHz, CDCl_3_): δ = 7.74 (m, 2H), 7.23 (m, 1H), 7.09 (m, 2H), 7.00 (m, 2H), 6.90 (m, 1H), 6.47 (br. s, 1H), 5.34 (d, 1H, *J* = 8.0 Hz), 3.99 (m, 1H), 3.83 (s, 3H), 3.27 (m, 1H), 2.80 (m, 1H), 2.22–2.12 (m, 2H), 1.92–1.65 (m, 8H), 1.54–1.49 (m, 2H), 1.40–0.83 (m, 14H) ppm; ^13^C NMR (100 MHz, CDCl_3_): δ = 188.4, 170.3, 166.0, 163.6, 154.7, 152.3, 142.0, 132.0, 130.6 (2C), 130.3, 125.9, 122.2, 121.5, 120.0, 114.2 (2C), 108.5, 83.2, 55.5, 53.8, 52.3, 47.0, 30.5, 30.0, 29.0 (2C), 26.4, 26.0, 25.9, 25.9, 25.2, 25.2, 22.8, 22.3 ppm. IR (mineral oil): 3183, 3060, 1783, 1760, 1739, 1698 cm^−1^. Anal. Calcd (%) for C_35_H_42_N_4_O_6_: C 68.38; H 6.89; N 9.11. Found: C 68.46; H 6.92; N 9.08. MS (ESI+): *m*/*z* calcd for C_35_H_42_N_4_O_6_+H^+^: 615.32 [M + H^+^]; found: 615.30.

*9-(4-Chlorobenzoyl)-1,3-dicyclohexyl-6-(2-hydroxyphenyl)-8-(isopropylamino)-1,3,6-triazaspiro[4.4]non-8-ene-2,4,7-trione***(3k).** Yield: 285 mg (92%); white solid; mp 151–153 °C; ^1^H NMR (400 MHz, CDCl_3_): δ = 7.67 (m, 2H), 7.47 (m, 2H), 7.23 (m, 1H), 7.06 (m, 1H), 7.01 (m, 1H), 6.90 (m, 1H), 6.45 (br. s, 1H), 5.67 (d, 1H, *J* = 4.0 Hz), 3.96 (m, 1H), 3.29 (m, 1H), 2.80 (m, 1H), 2.21–2.07 (m, 2H), 1.95–1.66 (m, 7H), 1.59–1.51 (m, 2H), 1.39–0.85 (m, 15H) ppm; ^13^C NMR (100 MHz, CDCl_3_): δ = 188.3, 170.0, 165.4, 154.5, 152.3, 139.3, 137.8, 130.5, 129.5 (2C), 129.3 (2C), 125.9, 122.0, 121.5, 119.9, 107.4, 83.0, 53.8, 52.3, 47.22, 30.5, 30.1, 29.0, 29.0, 26.4, 26.0, 25.9, 25.8, 25.2, 25.2, 22.7, 22.3 ppm. IR (mineral oil): 3281, 3080, 1774, 1721, 1704 cm^−1^. Anal. Calcd (%) for C_34_H_39_ClN_4_O_5_: C 65.96; H 6.35; N 9.05. Found: C 66.04; H 6.30; N 9.11. MS (ESI+): *m*/*z* calcd for C_34_H_39_ClN_4_O_5_+H^+^: 619.27 [M + H^+^]; found: 619.22.

*9-Benzoyl-1,3-dicyclohexyl-6-(2-hydroxyphenyl)-8-(phenylamino)-1,3,6-triazaspiro[4.4]non-8-ene-2,4,7-trione***(3l).** Yield: 285 mg (92%); yellow solid; mp 187–189 °C; ^1^H NMR (400 MHz, DMSO-*d*_6_): δ = 9.86 (s, 1H), 9.31 (s, 1H), 7.35 (m, 2H), 7.31–7.23 (m, 2H), 7.17 (m, 2H), 7.02 (m, 1H), 6.97–6.89 (m, 3H), 6.84 (m, 1H), 6.74 (m, 3H), 3.89 (m, 1H), 3.08 (m, 1H), 2.20–2.02 (m, 2H), 1.88–0.85 (m, 18H) ppm; ^13^C NMR (100 MHz, DMSO-*d*_6_): δ = 189.7, 169.8, 163.8, 154.1, 153.8, 141.8, 140.7, 138.3, 132.1, 129.9, 128.5 (2C), 128.0, 127.8 (2C), 126.2, 123.6, 121.3 (2C), 120.4, 119.0, 116.8, 113.8, 108.5, 82.3, 52.0, 51.0, 30.0, 29.4, 28.8, 28.7, 25.7, 25.3, 25.2, 24.9, 24.8 (2C) ppm. IR (mineral oil): 3398, 3247, 1768, 1723, 1704 cm^−1^. Anal. Calcd (%) for C_37_H_38_N_4_O_5_: C 71.83; H 6.19; N 9.06. Found: C 72.15; H 6.07; N 9.10. MS (ESI+): *m*/*z* calcd for C_37_H_38_N_4_O_5_+H^+^: 619.29 [M + H^+^]; found: 619.27.

*1,3-Dicyclohexyl-9-(4-ethoxybenzoyl)-6-(2-hydroxyphenyl)-8-(phenylamino)-1,3,6-triazaspiro[4.4]non-8-ene-2,4,7-trione***(3m).** Yield: 312 mg (94%); yellow solid; mp 187–189 °C; ^1^H NMR (400 MHz, DMSO-*d*_6_): δ = 9.85 (s, 1H), 9.20 (s, 1H), 7.34 (m, 2H), 7.24 (m, 1H), 7.01 (m, 1H), 6.93 (m, 3H), 6.83 (m, 1H), 6.75 (m, 3H), 6.66 (m, 2H), 4.09 (q, *J* = 6.8 Hz, 2H), 3.88 (m, 1H), 3.05 (m, 1H), 2.20–2.02 (m, 2H), 1.88–0.85 (m, 21H) ppm; ^13^C NMR (100 MHz, DMSO-*d*_6_): δ = 188.2, 169.9, 163.9, 161.7, 154.1, 153.8, 140.7, 140.5, 131.1, 130.3 (2C), 129.9, 128.2 (2C), 126.2, 123.4, 121.2 (2C), 120.5, 118.9, 116.7, 113.7 (2C), 109.2, 82.3, 63.2, 52.0, 51.0, 30.0, 29.5, 28.8, 28.7, 25.7, 25.3, 25.2, 25.0, 24.8, 24.8, 14.3 ppm. IR (mineral oil): 3354, 3250, 1771, 1720, 1705 cm^−1^. Anal. Calcd (%) for C_39_H_42_N_4_O_6_: C 70.68; H 6.39; N 8.45. Found: C 70.89; H 6.43; N 9.01. MS (ESI+): *m*/*z* calcd for C_39_H_42_N_4_O_6_+H^+^: 663.32 [M + H^+^]; found: 663.28.

*8-(Butylamino)-1,3-dicyclohexyl-9-(4-ethoxybenzoyl)-6-(2-hydroxyphenyl)-1,3,6-triazaspiro[4.4]non-8-ene-2,4,7-trione***(3n).** Yield: 270 mg (84%) (purity 90%); white solid; mp 130–132 °C; ^1^H NMR (400 MHz, DMSO-*d*_6_): δ = 9.81 (s, 1H), 7.64 (m, 2H), 7.22 (m, 2H), 7.13–6.77 (m, 10H), 4.16–4.06 (m, 3H), 3.83 (m, 1H), 3.15 (m, 1H), 2.98 (m, 2H), 2.55 (m, 1H), 2.40 (m, 1H), 2.15–1.96 (m, 3H), 1.87–0.67 (m, 18H) ppm; ^13^C NMR (100 MHz, DMSO-*d*_6_): δ = 188.4, 170.3, 163.7, 162.1, 154.1, 153.8, 143.9, 132.5, 130.4 (2C), 129.9, 126.2, 120.6, 118.9, 116.7, 114.3 (2C), 105.0, 82.4, 63.5, 51.9, 50.8, 45.5, 40.6, 32.2, 30.5, 30.2, 29.5, 28.8, 25.8, 25.4, 24.9, 19.2, 19.1, 14.4, 13.6, 13.3 ppm. IR (mineral oil): 3325, 3175, 1773, 1718, 1642 cm^−1^. Anal. Calcd (%) for C_37_H_46_N_4_O_6_: C 69.14; H 7.21; N 8.72. Found: C 68.89; H 7.25; N 8.71. MS (ESI+): *m*/*z* calcd for C_37_H_46_N_4_O_6_+H^+^: 643.35 [M + H^+^]; found: 643.30.

## 4. Conclusions

In conclusion, we developed a novel approach to 5-spiro-substituted 3-amino-1,5-dihydro-2*H*-pyrrol-2-ones based on the thermal decomposition of 1,3-disubstituted urea derivatives of 5-spiro-substituted 3-hydroxy-1,5-dihydro-2*H*-pyrrol-2-ones, which were readily prepared by their reaction with carbodiimides.

## Data Availability

The presented data are available in this article.

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
