# Peer review of "Amination of 5-Spiro-Substituted 3-Hydroxy-1,5-dihydro-2*H*-pyrrol-2-ones"

_molecules, 2021, doi:10.3390/molecules26237179_

Round 1
Reviewer 1 Report
The manuscript describes amination of 5-spiro-substituted 3-hydroxy-1,5-dihydro-2H-pyrrol-2-ones. The approach employed by the authors appears to work well. The paper is well written, and I recommend publication in Molecules after the corrections and modifications listed below.
The authors should present general Scheme for the synthesis of compounds 1a-g with the isolated yields (the yields should be presented in Scheme or in a separate Table).
Scheme 7. The label 3a should be replaced with 1a. The authors should present general Scheme for the synthesis of compounds 4a-n with the isolated yields (the yields should be presented in Scheme or in a separate Table).
For all synthesized compounds MS or HRMS data should be provided.
Author Response
- General Scheme for the synthesis of compounds 1a-g was included to the Scheme 3 of the manuscript (the yields are presented at Scheme 3).
- The label on Scheme 7 was corrected.
- MS data were added to Section 3.2.
Reviewer 2 Report
Khramtsova et al. reported their attempt to synthesize a series of 3-amino-1,5-dihydro-2H-pyrrol-2-ones by treating the corresponding 3-hydroxy analogues with carbodiimides follow by thermal decomposition of the generated ureas. We find the study well-designed and rationalized. The authors were able to place a reasonable hypothesis on the inactivity of the OH group and propose a facile solution. Therefore, we recommend publication of this manuscript after the following questions are satisfactorily addressed:
- We believe the benzoyl group (R4) can play a role in facilitate the hydroxyl group substitution. Have the authors tried the reaction with such substrate that bears no resonantly withdrawing group?
- The identity 3a was given for 2 different structures. Please assign each structure a separate ID?
- The obtained compounds showed no antimicrobial activity. It is known that unprotected nitrogen sites (N-H) can be active interacting sites. Those nitrogen atoms in all the products in this study were all masked with alkyl groups. Is it the reason why no activity was observed? In line with that, what do the authors propose to enhance the activity of these compounds?
- In the thermal decomposition step, the authors only used hexanes to wash the products. For those reactions with lower yields (<92%), what is the rest of the materials? Is it the unchanged starting material? If so, how could hexane dissolve the starting materials but not the products?
Author Response
- Unfortunately, substrates 2 bearing no resonantly withdrawing groups at C3 position are not available. They are not reported in the literature, since no one has succeeded to synthesize such compounds yet. Our research group performed many attempts to synthesize them, but no of them had any good results. If we succeed to prepare substrates 2 bearing no resonantly withdrawing groups at C3 position, it will be a subject for an individual publication.
- This misprint was fixed (The label on Scheme 7 was corrected).
- The presence of unprotected nitrogen sites (N-H) in the structure does not guarantee any antimicrobial activity. We tested antimicrobial activity in terms of international free screening program, which is suitable for rapid assessment of potency to create new antibiotics from unprecedented structures. We have no goal to create new antimicrobials on the basis of these compounds. In our future research, we need such structures to develop target-oriented molecules for several biological studies. This information is confidential and cannot be added to the present manuscript.
- In this transformation, lower yields are connected to the performance of isolation (filtration) and are not connected to solubility of side reactions (discussed in details in paper by M. Wernerova and T. Hudlicky DOI 10.1055/s-0030-1259018). According to UPLC-UV-MS data, the yields of the thermal decomposition step higher than 98%. Our manuscript gives real values of yields obtained in lab. Possibly, it should be discussed publicly to provide both isolated and UPLC/GC yields of the synthetic products.
Reviewer 3 Report
This manuscript describes the synthesis of some spirosubstituted compounds. I think this method might be useful for synthesizing similar analogues. Therefore, I think this manuscript is acceptable as an original paper in Molecules after considering the following comment.
The yields of compounds 1 and 4 are only shown in Material Methods. I think they could be shown in the main text as Tables, or in Table 2 together.
Author Response
- The yields of compounds 1 and 4 were added to Scheme 3 and Scheme 6.